# The Double-Edged Proteins in Cancer Proteomes and the Generation of Induced Tumor-Suppressing Cells (iTSCs)

**DOI:** 10.3390/proteomes11010005

**Published:** 2023-01-18

**Authors:** Kexin Li, Qingji Huo, Bai-Yan Li, Hiroki Yokota

**Affiliations:** 1Department of Pharmacology, School of Pharmacy, Harbin Medical University, Harbin 150081, China; 2Department of Biomedical Engineering, Indiana University Purdue University Indianapolis, Indianapolis, IN 46202, USA; 3Indiana Center for Musculoskeletal Health, Indiana University School of Medicine, Indianapolis, IN 46202, USA; 4Indiana University Simon Comprehensive Cancer Center, Indianapolis, IN 46202, USA

**Keywords:** iTSCs, induced tumor-suppressing cells, proteomes, conditioned medium

## Abstract

Unlike a prevalent expectation that tumor cells secrete tumor-promoting proteins and stimulate the progression of neighboring tumor cells, accumulating evidence indicates that the role of tumor-secreted proteins is double-edged and context-dependent. Some of the oncogenic proteins in the cytoplasm and cell membranes, which are considered to promote the proliferation and migration of tumor cells, may inversely act as tumor-suppressing proteins in the extracellular domain. Furthermore, the action of tumor-secreted proteins by aggressive “super-fit” tumor cells can be different from those derived from “less-fit” tumor cells. Tumor cells that are exposed to chemotherapeutic agents could alter their secretory proteomes. Super-fit tumor cells tend to secrete tumor-suppressing proteins, while less-fit or chemotherapeutic agent-treated tumor cells may secrete tumor-promotive proteomes. Interestingly, proteomes derived from nontumor cells such as mesenchymal stem cells and peripheral blood mononuclear cells mostly share common features with tumor cell-derived proteomes in response to certain signals. This review introduces the double-sided functions of tumor-secreted proteins and describes the proposed underlying mechanism, which would possibly be based on cell competition.

## 1. Introduction

Proteome-based characterization of varying types of cancer is an emerging area of cancer research beyond genome-wide DNA and transcriptome analyses [1]. The proteomic analysis may assist in the classification of poorly defined subtypes of cancers and the prediction of a potential new class of tumor-specific targets. The integration of mass spectrometry-based proteomics with next-generation DNA and RNA sequencing profiles can deepen our understanding of the role of post-translational modifications such as protein phosphorylation and acetylation [2]. Pancreatic ductal adenocarcinoma (PDAC), for instance, is a highly aggressive cancer with poor patient survival [3,4]. Toward understanding the underlying molecular alterations that drive PDAC oncogenesis, proteomic analyses, including phosphoproteomics and glycoproteomics, have been used to characterize proteins and their modifications [5]. In osteosarcoma, an aggressive bone tumor with a high metastasis rate in the lungs [6,7], proteomics is conducted to identify differences in the functional network including chaperones, structural proteins, stress-related proteins, proteins of the glycolysis/gluconeogenesis pathway, and oxidoreductases in the two and three-dimensional culture systems [8]. One of the main purposes of characterizing cancer proteomes is to identify tumorigenic and antitumorigenic proteins and develop a protein-based strategy for cancer treatment. This review focuses on the recent advance in our understanding of cancer proteomes as well as noncancer proteomes, and evaluates their unconventional involvement in the progression and suppression of cancer growth.

Before reviewing the role of cancer proteomes, we will briefly describe the linkage of oncogenes to cancer proteomes. An oncogene is a mutated gene that has the potential to induce oncogenesis and converts normal cells into cancerous cells [9]. Prior to genetic alterations such as an increase in the copy number, mutations, chromosomal translocations, and epigenetic modifications, an oncogene is called a proto-oncogene [10]. Cellular myelocytomatosis (cMyc) and the Kirsten rat sarcoma viral oncogene homolog (K-Ras), for instance, are two of the well-known proto-oncogenes [11,12]. cMyc is a transcription factor that is involved in cell proliferation, transformation, and apoptosis [13,14]. K-Ras is a GTPase that regulates various cellular signalings for cell cycles, proliferation, and migration, including phosphoinositide 3-kinase (PI3K) and extracellular signal-regulated kinase (ERK) [15,16,17]. In many types of cancer, these proto-oncogenes are dysregulated and expression profiles of many other proteins are significantly altered [18,19]. In about 90% of PDAC cases, K-Ras is mutated and the mutated K-Ras drives tumorigenic signaling cascades [20]. Thus, most efforts have been directed to inhibit the action of oncogenes [21,22,23,24], for instance, the development of selective K-Ras inhibitors [25]. In some of the recent studies, however, the other side of the oncogene actions has been revealed [26,27]. Notably, some oncogenic proteins such as TGFβ [28], spleen tyrosine kinase [29], and Sirtuin [30] may induce antiproliferative and proapoptotic signaling in a context-dependent fashion [31]. Regarding cMyc, its deregulated expression occurs in many cancers and is often associated with poor prognosis [32]. At the same time, its role is not single-sided, and it is unclear whether cMyc is instrumental in the tumor initiation and progression, and whether its inactivation would lead to tumor regression [32]. The observed double-sided nature of oncogenes, as well as the cancer cell-derived proteomes [33,34], are reviewed herein, which is critically important in developing an effective therapeutic strategy and inhibiting cancer progression.

While oncogenes by definition stimulate many cancerous features, their effects on neighboring cells via extracellular proteomes can be counterintuitive. An intriguing observation is the counterintuitive effects of cMyc on neighboring cells. In Drosophila, cells overexpressing d-myc, which is a homolog of cMyc, are reported to eliminate neighboring wild-type cells [35]. It is also reported that the conditioned medium (CM), derived from dmyc-overexpressing cells, induces cell death of nearby wild-type cells [36]. We also reported that cMyc-overexpressing cancer cells generate a tumor-suppressing CM that induces apoptosis to cohort tumor cells [36]. Consistently, tumor-suppressing CM, derived from cMyc-overexpressing cells, is shown to suppress metabolic activities, proliferation, two-dimensional motility, and transwell invasion of multiple breast cancer cell lines [37]. Furthermore, the systemic administration of tumor-suppressing CM to a mouse model of breast cancer and bone metastasis is reported to inhibit the growth of mammary tumors and the osteolytic destruction of tumor-invaded bone [37,38,39]. An emerging paradigm is that some of the oncogenes have a double-edged role as a tumor enhancer as well as a tumor suppressor [40,41,42]. Although paradoxical, a provocative question is whether a novel therapeutic strategy could be developed not by inhibiting those proto-oncogenes but by utilizing their neighbor-removing capabilities.

The concept of induced tumor-suppressing cells (iTSCs) was introduced by our research group in 2021 [37,38,39,43,44,45,46,47]. By definition, iTSCs suppress the progression of tumor cells and they generate tumor-suppressing proteomes in their secreted CM. Tumor-suppressing CM is enriched with a group of tumor-suppressing proteins, some of which have been known as tumorigenic proteins. Interestingly, iTSCs can be generated using two of the four transcription factors, which are employed in creating induced pluripotent stem cells (iPSCs) [39]. This article reviewed the brief history of iTSCs, the procedure of generating tumor-suppressing proteomes, and their possible therapeutic use in cancer treatments.

## 2. Induced Tumor-Suppressing Cells (iTSCs) and Their Conditioned Medium (CM)

The first generation of iTSCs is induced by the activation of Wnt signaling in osteocytes, the most abundant type of bone cells in the calcified matrix [43]. Wnt signaling is activated in response to mechanical stimulation of the bone, and it promotes loading-driven bone formation [48]. Wnt signaling is also involved in varying tumorigenic actions in many types of cancer [49], and its inhibition has been the main target for primary and metastatic cancer [50]. Paradoxically, however, the activation of Wnt signaling by the overexpression of Lrp5, a Wnt co-receptor [51], and β-catenin in osteocytes [52], generated osteocyte-derived tumor-suppressive CM. The proliferation, two-dimensional motility, and transwell invasion of breast cancer cell lines such as MDA-MB-231 were suppressed in response to the treatment with Wnt-activated osteocyte-derived CM [43]. Subsequently, the treatment of osteocytes and other bone cells such as mesenchymal stem cells (MSCs) with a pharmacological Wnt activator, BML284, was shown to generate iTSCs and tumor-suppressive CM [43,44].

Besides the activation of Wnt signaling, the activation of PI3K/Akt [53] and protein kinase A (PKA) signaling [54] was shown to generate tumor-suppressing CM from osteocytes, osteoblasts, osteoclasts, and MSCs [38,43,44,46] (Table 1). PKA signaling is also known as a cAMP-dependent kinase pathway, and it is involved in the regulation of glycogen, sugar, and lipid metabolism [55]. PKA signaling is implicated in the initiation and progression of many types of tumors and is proposed as a biomarker for cancer detection and as a potential target for cancer therapy [56]. Besides the above bone-linked adherent cells, PKA-activated iTSCs were generated from cells in suspension such as peripheral blood mononuclear cells (PBMCs), lymphocytes, and monocytes [57]. CM retained its antitumor capability regardless of nuclease digestion to remove DNA and RNA, and filtration to eliminate small molecules such as amino acids, metabolites, and neurotransmitters [46]. Therefore, it is considered that the main antitumor action of CM is caused by the secreted proteomes from iTSC-generating cells.

In addition to the generation of iTSCs from nontumor cells, cancer cells and isolated cancer tissues were utilized to generate tumor-suppressive CM. For instance, breast cancer cell lines (MDA-MB-231, MDA-MB-436, MCF-7, EO771, and 4T1.2), prostate cancer cell lines (PC-3 and TRAMP-C2ras), PDAC cell lines (PANC1), and osteosarcoma cell lines (MG63, U2OS) were converted into iTSCs by regulating varying signaling pathways and overexpressing pro-oncogenic genes [37]. Besides Lrp5 and β-catenin, the overexpression of proto-oncogenic genes such as cMyc, octamer-binding transcription factor 4 (Oct4), and zinc finger protein SNAI1 (Snail) was shown to generate iTSCs [38,39,44]. Of note, cMyc and Oct 4 are two of the four transcription factors, i.e., Yamanaka factors, to generate iPSCs [58], while Snail is one of the key transcription factors that are involved in an epithelial-to-mesenchymal transition [59]. Collectively, available data indicate that iTSCs are generated in a counterintuitive procedure, in which the tumorigenic pathways are stimulated in tumor and nontumor cells.

The generation of iTSCs was achieved not only by the activation of tumorigenic signaling but also by the inactivation of antitumorigenic signaling such as an AMP-activated protein kinase (AMPK) pathway [60]. Although the pro- and antitumorigenic role of AMPK signaling could depend on cellular conditions [61], its inactivation by Dorsomorphin, a pharmacological AMPK inhibitor, converted PBMCs into iTSCs and generated PBMC-derived tumor-suppressing CM. In addition to MSCs, PBMCs can be harvested from cancer patients and autologous MSC/PBMC-derived CM can be synthesized. Taken together, it is possible to engineer iTSCs from varying tumor and nontumor cells by regulating tumorigenic signaling pathways. Of note, the efficacy of tumor-suppressing capabilities differs among iTSC-deriving host cells and the pathways to be regulated (Figure 1). For instance, to generate PBMC-derived tumor-suppressive CM, the activation of PKA signaling and the inactivation of AMPK signaling are effective. However, the activation of Wnt signaling is less effective for converting PBMCs than MSCs into iTSCs. Further studies are recommended to develop the most effective procedure to generate potent antitumor CM from various types of iTSC host cells.

## 3. Double-Edged Role of Tumor-Suppressing Proteins

Mass spectrometry-based global proteomics analyses predicted the tumor-suppressing protein candidates, which were enriched in tumor-suppressive CM. The candidate proteins included Enolase 1 (Eno1) [62], Moesin (MSN) [63], Calreticulin (Calr) [64], Ubiquitin C (Ubc) [65], Histone H4 (H4) [66], Heat shock protein 90 alpha family class B member 1 (Hsp90ab1) [67], etc. (Table 2). In vitro analyses using recombinant proteins as well as their gain and loss-of-functions by plasmid transfection and RNA silencing revealed that most of the predicted tumor-suppressing proteins in CM acted as oncoproteins inside tumor cells and tumor suppressors in the extracellular domain, although Calr is known as a tumor suppressor intracellularly.

Unraveling the mechanism of iTSC-driven tumor suppression requires intensive proteome and protein interaction analyses (Figure 2), but several regulatory axes have been proposed through publicly available protein databases and protein interaction analyses. For instance, extracellular Eno1 recombinant proteins are reported to suppress the metabolic activities of breast cancer cells and act as cytotoxic agents by downregulating Snail, TGFβ, and MMP9 [39], and its antitumor action was shown to be mediated by the Eno1-CD44 regulatory axis. CD44 is a cell surface adhesion receptor [39], and is known as a cancer stem cell marker in several cancer cells [68]. By contrast, the overexpression of Eno1 in breast cancer cells upregulated the above tumorigenic genes and elevated their proliferation and transwell invasion. Another example is MSN, which is a member of the ezrin-radixin-moesin (ERM) family of proteins. These three proteins in the family are known to cross-link the plasma membrane with the actin cytoskeleton [69]. An accumulating body of evidence indicates that ERM proteins regulate cellular signaling implicated in cell motility and adhesion, which are involved in the proliferation and migration of cancer cells [70]. It is shown that the antitumor action of extracellular MSN is mediated by CD44 and fibronectin 1 (FN1) [38]. FN1 plays a major role in cell adhesion and migration, and its elevated expression is reported in many types of cancer [71]. It is reported that FN1-expressing MSCs promote breast cancer metastasis [72].

Calr (aka Calregulin) is a calcium-binding protein that is mostly localized in the endoplasmic reticulum [73]. It functions as a calcium buffer and a chaperone in normal cells, but its low level is associated with accrued malignant features [64]. Unlike atypical tumor-suppressing proteins that present a double-sided role, the high expression level of Calr inversely correlates with patient survival [74]. Calr is reported to interact with CD47, which is an integrin-associated transmembrane protein [75]. CD47 is known as an immunoregulator, and its high expression in cancer cells blocks the induction of immune responses [76]. Interestingly, extracellular histones (H1, H2A, H2B, H3, and H4) have been reported to play multiple roles in various diseases [77,78,79,80]. Histone H4 is one of the five core histone proteins that are involved in the formation of nucleosomes in chromatin [81]. The unexpected cytotoxic role of extracellular H4 is reported in sepsis via the interaction with toll-like receptor (TLR) 2 and 4, which are upregulated in inflammatory injuries [82]. Regarding Hsp90ab1 (aka Hsp90b), it is shown in osteosarcoma cells that secreted Hsp90ab1 proteins inhibit the activation of latent TGFβ [83]. In the bone-tumor microenvironment, TGFβ is considered critical for driving the feedforward vicious cycle of tumor growth [84]. It is noted, however, that it has a conflicting role as a tumor suppressor [85] as well as an enhancer, which may present a challenge in developing TGFβ-related therapeutics [84].

So far, regulatory mechanisms have been proposed for several tumor-suppressing proteins (Table 2), whereas the overall CM’s mechanism of action is yet to be elucidated. No clear mechanism is presented for the antitumor actions of Ubc, peptidyl-prolyl cis-trans isomerase B (Ppib), eukaryotic translation elongation factor 2 (Eef2), and Vinculin (VCL). Interestingly, these three proteins are generally treated as tumor enhancers in various cancers. Ubc is a source of ubiquitin that contributes to ubiquitylation for protein degradation. While the direct role of Ubc in cancer progression is not reported, ubiquitin-mediated protein degradation can stabilize oncoproteins and increase tumor suppressors, contributing to tumorigenesis and cancer progression [86]. PPib is localized in the endoplasmic reticulum for various cellular functions, including inflammation, apoptosis, and mitochondrial metabolism, and it is known to be involved in diseases such as ischemia, AIDS, and cancer [87]. Eef2 is reported as a potential biomarker of prostate cancer [88] and a promoter of the proliferation of ovarian cancer [89]. VCL is a membrane protein in focal adhesion and it links integrin with the actin cytoskeleton [90]. It is reported to orchestrate the progression of prostate cancer [91].

**Table 2 proteomes-11-00005-t002:** List of atypical tumor-suppressing proteins.

Symbol	Name	Predicted Antitumor Action	Reference
Eno1	Enolase 1	Interact with CD44	[37,39,46]
MSN	Moesin	Interact with CD44 and fibronectin 1 (FN1)	[37,38,39,46]
Calr	Calreticulin	Interact with CD47	[44,57]
Ubc	Ubiquitin C	Unknown	[37]
H4	Histone H4	Interact with TLR2/4	[47,92]
Hsp90ab1	Heat shock protein 90 alpha family class B member 1	Immunoprecipitates latent TGFβ and inactivate TGFβ	[38,39,44,45,46]
Ppib	Peptidylprolyl isomerase B	Unknown	[44]
Eef2	Eukaryotic elongation factor 2	Unknown	[39]
VCL	Vinculin	Unknown	[39]

In addition to in vitro assays that evaluate the efficacy and mechanism of antitumor actions, preclinical studies using a mouse model supported the in vivo efficacy of proteome-based therapy. For instance, the systemic administration of Lrp5-overexpressing CM reduced the weight of mammary tumors and the size of tumor-invaded areas in the lung [38]. Moreover, the progression of primary bone tumors (osteosarcoma), as well as secondary bone tumors (breast cancer-associated bone metastasis), was suppressed by the administration of iTSC-derived CM in mouse models [38,39,44,57]. The engineered CM was also applied to freshly isolated human breast cancer tissues, and the shrinkage of cancer tissue fragments was observed [37,38,39,44]. The viability of patient-derived xenografted osteosarcoma tissues was also suppressed by PKA-activated PBMC-derived CM that was prepared by the peripheral blood samples of healthy humans [57].

One major question, which is linked to potential side effects, is whether the cytotoxic effects of iTSC-derived CM present tumor selectivity. To minimize undesirable side effects, the cytotoxic action of CM, which induces proapoptotic signaling, should be selective to tumor cells, and ideally, nontumor cells would receive no or significantly smaller effects than tumor cells. In a study of bone metastasis, the cytotoxic effect of iTSC-derived CM was compared between breast cancer cells and noncancer cells such as osteoblasts, osteocytes, and MSCs [38,43,44,45,47]. Available data indicate that tumor-suppressive CM and the selected tumor-suppressing proteins in general induced stronger cytotoxic effects on tumor cells than nontumor cells. The result is probably linked to interactions of tumor-suppressing proteins with cell-surface proteins of tumor cells. For instance, the expression level of CD44, which may interact with Eno1 and MSN, can be expressed higher in cancer cells than noncancer cells, and thus the impact of CD44-mediated cytotoxicity is stronger in tumor cells. In preclinical studies using mouse models, the reduction in body weight during the administration of CM is in general insignificant compared to the effect with the administration of chemotherapeutic agents. Another major question is the effects of CM’s action on immune responses. It is important to evaluate the compatibility of CM’s action with immunotherapy such as CAR-T-cell therapy [93].

Another interesting question is about the form of intercellular communication between iTSCs and tumor cells. Although we have primarily discussed the possible binding of tumor-suppressing proteins to membrane proteins in this review, the tumor-suppressing effects may occur through the binding of proteins in the extracellular microenvironment, extracellular vesicles, and/or interactions with recently described nanoparticles such as exomes [94] and supermeres [95].

## 4. Cell Competition and Cooperation

An interesting question on tumor-suppressive CM is its significance in the context of Darwin’s evolutionary theory. Intuitively, cancer cells are considered to proliferate and migrate as a group by taking advantage of any resources. Thus, they may cooperate and help each other to maximize their growth as a group. However, the procedure to generate iTSCs provided an alternative view on the role of cell competition and cooperation among tumor and nontumor cells. A common perception of cancer progression emphasizes the role of cooperation in promoting tumor growth and metastases, in which tumor cells help enhance their mutual survival [96]. They may cooperatively interact directly via cell-surface proteins, and indirectly with secreted proteins and exosomes. By contrast, cell competition may favor the survival of the fittest at the expense of less-fit cells [97,98,99,100,101,102]. It is rational to assume that both cell cooperation and competition are at work among tumor and nontumor cells, and the role of cooperation and competition may differ depending on the context. At a first glance, it is paradoxical to generate tumor-suppressing proteomes by activating tumorigenic signaling or inactivating antitumorigenic signaling. From an evolutionary viewpoint, however, it is reasonable to presume that highly-fitted tumor cells become the ultimate winner by eliminating less-fitted cells. Notably, the behavior of super-fit cells can be the opposite of dying tumor cells, for instance, in response to chemotherapeutic agents. Less-fit cells may not survive, but they may induce tumor-promoting CM to increase the survival of neighboring tumor cells and contribute to making them fitter for survival [103]. It is also important to know the effects of cellular heterogeneity among individual cancer cells. For instance, significant variations are reported in single-cell analyses in the development of pancreatic injury and cancer [104,105]. The heterogeneous interactions of iTSCs with the microenvironment would alter the outcome of the survival-of-the-fittest evolutionary process.

## 5. Application

The application of iTSC-derived proteomes can be considered at several different levels, including the transplantation of iTSCs, the administration of CM, the application of the selected cocktail proteins that are enriched in CM, and the development of peptides that are critical in inducing antitumor actions in CM. First, the transplantable iTSCs can be synthesized using MSCs [106], PBMCs, and lymphocytes [107] that are autologous to cancer patients. MSCs can be harvested from the bone marrow and adipose tissues. Engineering lymphocytes may allow for the use of technology analogous to that of CAR-T cells for immunotherapy [108,109]. Autologous cell-based therapy is experimentally and clinically practiced in regenerative medicine, in the area of tissue repair, hair regrowth, etc., including an application of iPSCs [110,111]. Second, the administration of CM is expected to have an advantage over the single or several selected proteins or peptides because of its large number of versatile components. Stem-cell secretome therapy is experimentally applied by a biotech firm such as Anova in Europe as regenerative medicine with a broad spectrum of treatment possibilities, including multiple sclerosis, osteoarthritis, rheumatoid arthritis, Parkinson’s disease, motor neuron disease, spinal cord injury, and stroke. The advantage of an integrative mixture of multiple tumor-suppressing proteins may become a hurdle in gaining FDA approval for clinical applications. Nevertheless, a proteome-based therapy can provide a comprehensive set of substances that are based on Mother Nature’s strategy to eliminate cancer cells. Another approach is to select a group of effective proteins and peptides and apply them as a cocktail. It is not known, however, whether iTSC-derived CM contains peptides that present potent tumor-suppressing capabilities. Available data indicate that proteins or peptides, smaller than 3 kD, are not the major source of antitumor actions of iTSC-derived CM. The cocktail can be custom-designed based on the diagnosis of individual cancer patients. Lastly, it is necessary to test whether CM or cocktail proteins could be used with standard-of-care chemotherapeutic agents such as Paclitaxel [112] and Cisplatin [113].

## 6. Perspective

To generate iTSCs, we employ pharmacological agents that activate tumorigenic signaling or inactivate it. The procedure is counterintuitive and opposite to regular uses of pharmacological agents. To improve tumor-suppressing capabilities, it is important to employ highly effective agents, which are suitable for iTSC-generating host cell types. We also need to evaluate a proper matching of pharmacological agents with iTSC-generating host cells. One caution is the potentially detrimental responses of pharmacological agents to the survival of iTSCs. A higher concentration of CW008 (PKA activator), for instance, tends to reduce the rate of cellular proliferation of iTSCs. The generation of iTSCs requires the activation of tumorigenic signaling, and currently, the availability of those protumorigenic agents is limited. It is not clear whether the observed inhibitory effect on the growth of iTSCs is caused by the oncogenic activation itself or nonspecific side effects. The most effective combination of the agent and iTSC-generating cells may differ depending on the target cancer type, such as breast cancer, prostate cancer, pancreatic cancer, osteosarcoma, etc. Although the current efforts are mainly focused on breast cancer, prostate cancer, and pancreatic cancer, it is recommended to test whether iTSC-derived CM might exert antitumor actions for other cancer types such as blood cancer, lung cancer, liver cancer, glioblastoma, etc.

In cell-based therapy, the discovery of iPSC technology has opened up unprecedented opportunities in regenerative medicine, gene therapy, drug discovery, and disease modeling [114,115]. Ongoing autologous iPSC-derived cell regeneration includes retinal pigment epithelial cells in the eye, dopaminergic progenitor cells and neurons in the brain, myogenic stem cells for the treatment of muscular dystrophies, and neural crest-derived dermal papillae for hair loss [111]. While the use of iPSC technology for cancer immunotherapy is considered [116], iPSCs with an incomplete reprogramming process by the four transcription factors may induce tumorigenesis [117]. It is imperative to make sure that iTSCs and their CM do not induce any tumorigenic responses.

Lastly, it is expected that future cancer therapy, particularly for the treatment of advanced cancer, requires the application of multiple agents to cope with ever-changing cancer cells. Most agents may initially be effective, whereas cancer cells tend to develop resistance and the efficacy may not be sustained [118]. The advantage of proteomes and CM is that they include a spectrum of tumor-suppressing proteins, and cancer cells may not easily develop resistance. Below is a list of pressing questions that should be addressed to advance the current iTSC technology.

Generation of iTSCs: What determines the most effective procedure to generate iTSCs? This question is linked to the genes being overexpressed, signaling pathways to be regulated, and the compatibility of host cells with the genes and pathways.Variations among iTSCs: What is the advantage of using autologous MSCs and PBMCs as a host of iTSCs? Is there any advantage of generating iTSCs from patient-derived cancer cells?Target cancer types: Is iTSC-derived CM effective for all types of cancer? So far, in vitro and preclinical studies supported the efficacy for breast cancer, prostate cancer, pancreatic cancer, and osteosarcoma using cell lines, primary cells, and freshly isolated cancer tissues. Variations in efficacy were observed, however, and the question is how to enhance tumor-suppressive actions.Protein isoforms and modifications: Do protein isoforms and modifications such as phosphorylation alter the antitumor capability of atypical tumor-suppressing proteins? No existing studies have evaluated the role of differential splicing, post-translational modification, and DNA mutations.Nonprotein molecules: Do nonprotein molecules in iTSC CM contribute to tumor-suppressive capabilities? Neurotransmitters such as dopamine are shown to act as tumor suppressors, while metabolites such as cholesterol may act as a tumorigenic factor [119,120]. It is also shown that nucleic acids in exosomes affect tumor progression [121]. Further analyses are necessary to evaluate whether any nonprotein molecules significantly contribute to the antitumor action of iTSC CM.Mechanism of actions: Do atypical tumor-suppressing proteins exert their antitumor actions by interacting with free proteins, membrane-bound proteins, and extracellular proteins? While existing studies have been focused on interactions with plasma membrane-bound receptors, many other mechanisms can be considered, including the interaction with a tumor microenvironment. For instance, atypical tumor-suppressing proteins may interact with extracellular vesicles or the more recently described nanoparticles such as exomeres [94] and supermeres [95].

## 7. Limitation

This review has a few limitations. Cancer proteomes are composed of a complex group of proteins, and they dynamically interact with varying cells locally in a tumor microenvironment, as well as globally with distant tissues. Few existing studies, however, have been conducted to analyze the isoforms of tumor-suppressing proteins and their modifications, as well as proteoforms that are derived from variations in genomic sequences [122]. The proteoforms may differ depending on the hosting cell type of iTSCs. The efficacy of antitumor actions may depend on the varying protein species. Furthermore, some proteins in iTSC-derived CM are cell-membrane proteins and are not considered secretory proteins. Understanding a cellular process to be secreted should help decipher the double-edged role of tumor-suppressing proteins. Lastly, it is important to conduct pharmacokinetics analyses and evaluate the stability and availability of the described tumor-suppressing proteins in preclinical and clinical settings.

## 8. Conclusions

Focusing on the role of secretory proteomes of the tumor as well as nontumor cells, including MSCs and PBMCs, that can be collected from cancer patients, this review described the counterintuitive generation procedure of iTSCs, a novel type of tumor-fighting cells, which produce atypical tumor-suppressing proteins such as Eno1, MSN, Ubc, Hsp90ab1, etc. Most of those proteins act as oncogenic intracellularly and as antioncogenic extracellularly, except for Calr. The proposed regulatory mechanism is far from complete, and the function of each of those tumor-suppressing proteins may differ depending on the cancer type. The double-edged action of these proteins presents a new potential dimension in cancer treatments as cell and protein-based therapies. The existing in vitro, ex vivo, and preclinical mouse studies warrant further investigation of the antitumor actions and underlying mechanisms, and the development of translatable iTSC technologies.

## Figures and Tables

**Figure 1 proteomes-11-00005-f001:**
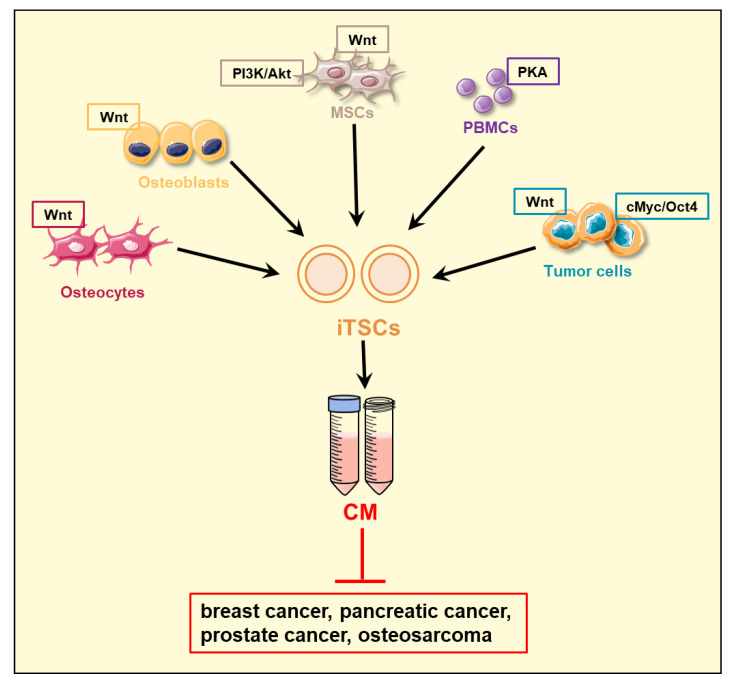
Generation of induced tumor-suppressing cells (iTSCs). iTSCs can be generated from osteocytes, osteoblasts, mesenchymal stem cells (MSCs), peripheral blood mononuclear cells (PBMCs), and varying types of tumor cells by the activation of Wnt, PI3K, and PKA signaling pathways, as well as the overexpression of cMyc and Oct4. The iTSC-derived conditioned medium (CM) has been shown to be effective in suppressing the progression of breast, pancreatic, and prostate cancer cells, as well as osteosarcoma cells.

**Figure 2 proteomes-11-00005-f002:**
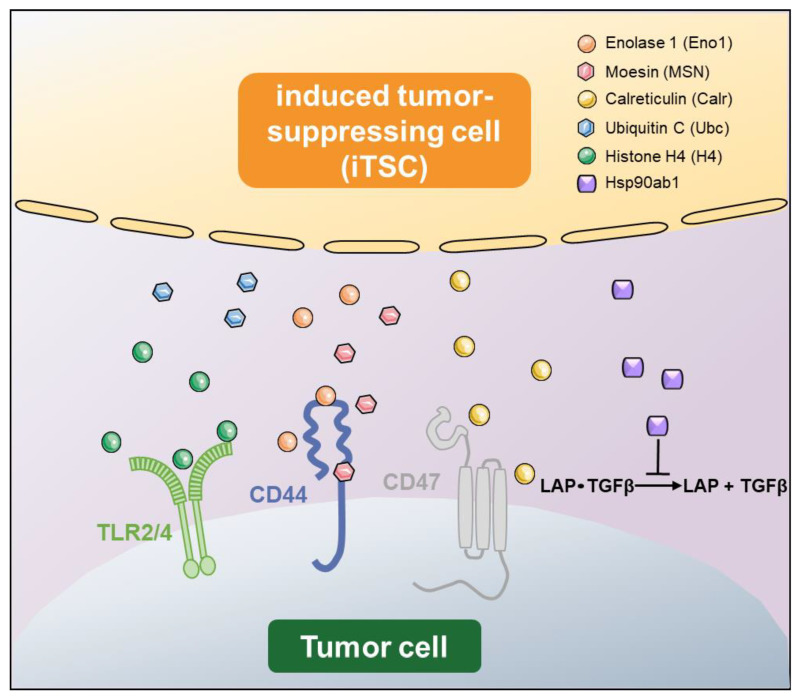
Proposed mechanism of tumor-suppressing actions by iTSC-derived conditioned medium (CM). iTSCs produce a spectrum of secretory proteins in their conditioned medium. They include Enolase 1 (Eno1), Moesin (MSN), Calreticulin (Calr), Ubiquitin C (Ubc), Histone H4 (H4), Hsp90ab1 (HSP), etc. It is proposed that most of those tumor-suppressing proteins interact with cell-surface proteins on tumor cells such as TLR2/4, CD44, and CD47, and induce antitumor actions.

**Table 1 proteomes-11-00005-t001:** List of the signalings to be regulated to make iTSCs from varying cells.

Signaling Regulation	Drug	iTSC-Generating Cells	Reference
PKA activation	CW008	lymphocytes, PBMCs	[57]
Wnt activation	BML284	MSCs, osteocytes osteoblasts, osteoclasts, tumor cells	[38,43,44,46]
PI3K/AKT activation	YS49	MSCs	[44,45]
cMyc overexpression		tumor cells	[39]

Note: PBMC = peripheral blood mononuclear cells, and MSC = mesenchymal stem cells.

## Data Availability

Not applicable.

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
