# Peer review of "The Double-Edged Proteins in Cancer Proteomes and the Generation of Induced Tumor-Suppressing Cells (iTSCs)"

_proteomes, 2023, doi:10.3390/proteomes11010005_

Round 1
Reviewer 2 Report
In this review, Li and colleagues described and summarized the counterintuitive double-edged nature of tumor-secreted proteins and the generation of induced tumor-suppressing cells (iTSCs). This review is well-organized with enriched information from observation, molecular mechanisms to clinical applications. It is easy to follow and I myself really enjoyed reading through it. I would suggest the authors make a list of important questions in the field that need to be tackled immediately in the section of Perspective.
Reviewer 3 Report
GENERAL COMMENTS
This review highlights the main findings on the topic of tumor suppressor genes and their paradoxical function in cancers, notably in the context of therapy resistance and plasticity. The topic is timely and interesting and is in my opinion highly citable. It is a pleasant and instructive read. Please see my comments.
MAJOR COMMENTS
Please add a section (likely between sections 3 and 4) on intercellular communication between iTSCs and tumor cells. Even though your paper discusses proteome, it would be interesting to see if you believe that these tumor suppressor effects occur through free proteins, protein binding to the microenvironment, extracellular vesicles, or the more recently described nanoparticles such as exomeres (PMID: 29459780) and supermeres (PMID: 34887515).
In section 4, can you add a comment on cellular heterogeneity in cancers. Single-cell studies have described significant variations between different populations of precursor lesions and invasive cancer. Notably, in Zhibo Ma et al. (PMID: 34695382) and Schlesinger et al. (PMID: 32908137) described that in response to injury and during the development of low-grade precursors of pancreatic cancers, there is significant cellular heterogeneity during development, with basal cells expressing CD44 while surface cells would express other gastric lineage markers. This would suggest some type of cellular diversity occurs, with likely survival-of-the-fittest occurring. Furthermore, this evolutionary process would occur in a spatially dependent fashion through likely interactions with the microenvironment and iTSCs.
MINOR COMMENTS
Line 106 put the symbol for beta.
